# Development of an Imaging Flow Cytometry Method for Fungal Cytological Profiling and Its Potential Application in Antifungal Drug Development

**DOI:** 10.3390/jof9070722

**Published:** 2023-06-30

**Authors:** Courtney L. McMahon, Marisol Esqueda, Jieh-Juen Yu, Gina Wall, Jesus A. Romo, Taissa Vila, Ashok Chaturvedi, Jose L. Lopez-Ribot, Floyd Wormley, Chiung-Yu Hung

**Affiliations:** Department of Molecular Microbiology and Immunology and South Texas Center for Emerging Infectious Diseases, The University of Texas at San Antonio, San Antonio, TX 78249, USA

**Keywords:** antifungal, drug discovery, cytological profiling, imaging flow cytometry, phenotypic screening

## Abstract

Automated imaging techniques have been in increasing demand for the more advanced analysis and efficient characterization of cellular phenotypes. The success of the image-based profiling method hinges on assays that can rapidly and simultaneously capture a wide range of phenotypic features. We have developed an automated image acquisition method for fungal cytological profiling (FCP) using an imaging flow cytometer that can objectively measure over 250 features of a single fungal cell. Fungal cells were labeled with calcofluor white and FM4-64FX, which bind to the cell wall and lipophilic membrane, respectively. Images of single cells were analyzed using IDEAS^®^ software. We first acquired FCPs of fungal cells treated with fluconazole, amphotericin B, and caspofungin, each with a distinct mode of action, to establish FCP databases of profiles associated with specific antifungal treatment. Once fully established, we investigated the potential application of this technique as a screening methodology to identify compounds with novel antifungal activity against *Candida albicans* and *Cryptococcus neoformans*. Altogether, we have developed a rapid, powerful, and novel image-profiling method for the phenotypic characterization of fungal cells, also with potential applications in antifungal drug development.

## 1. Introduction

Invasive fungal infections are a major cause of morbidity and mortality for patients who are immune-compromised or have undergone immunosuppressive treatment [1]. Fungi are eukaryotes and closely related to the host; thus, many compounds that are toxic to fungal pathogens are also toxic to humans, thereby restricting the number of antifungal drugs [2]. There are only four classes of antifungal agents: polyenes (e.g., amphotericin B), azoles (e.g., fluconazole, itraconazole, voriconazole, posaconazole, and isavuconazole), echinocandins (e.g., caspofungin, micafungin, and anidulafungin), and antimetabolites (e.g., 5-flucytosine) for clinical treatment [2,3]. Polyenes inhibit fungal growth by binding ergosterol in the fungal cell membrane, resulting in the alteration of membrane stability and permeability. Azoles act primarily by inhibiting fungal cytochrome P450 enzymes, such as 14α-lanosterol demethylase that plays an essential role in ergosterol biosynthesis. Echinocandins inhibit 1,3-β-D-glucan synthesis to weaken fungal cell walls and trigger cell lysis. Flucytosine is a pyrimidine analogue that selectively impedes fungal nucleic acid and protein synthesis. The efficacy of each antifungal to treat different mycoses varies. For example, the echinocandins have emerged as an important therapeutic option for candidiasis, but display little-to-no clinically useful activity against *Cryptococcus* or the mucorales [4,5]. Furthermore, due to toxicity associated with amphotericin B [6] and emerging clinical resistance to azoles and echinocandins [7,8], the choice of antifungals for the treatment of invasive mycoses is further constricted.

Effective antifungal drugs may inhibit fungal growth and alter fungal cytological profiles. A variety of phenotypical profiling methods have been shown to be powerful tools for characterizing the effect of compounds on target cells [9,10]. These phenotypic profiling techniques are typically conducted by analyzing fluorescence-labeled, phenotype-associated antigenic targets in cells of interest with an automated fluorescence microscope using a microplate format [11,12,13]. With the advances in imaging technology, the utilization of imaging flow cytometry (IFC) could further improve and enhance cytological profiling for many biomedical applications. The IFC method combines features of flow cytometry and fluorescent microscopy with advances in data-processing algorithms. This imaging flow technique can objectively measure over 250 characteristic features of a cell including size, shape, texture, nuclear DNA morphology, cell wall integrity, and membrane permeability, for an in-depth assessment of ae compound’s effect on the examined fungal pathogens. Here, we report the development of a novel automated image acquisition and processing method using IFC to characterize the effect of antifungal drugs. We also provide evidence for its potential utility in antifungal drug development, more specifically in the screening of compound libraries against two model pathogens, *Candida albicans* and *Cryptococcus neoformans*.

## 2. Materials and Methods

### 2.1. Fungal Strains and Culture Conditions

*C. albicans* strain SC5314 and *Cryptococcus neoformans* strain H99 (serotype A) were utilized for the majority of experiments, unless otherwise specified in the text. We also evaluated 3 other non-albicans *Candida* species, including *C. auris* (strains 0384 and 0390), *C. lusitaniae,* and *C. glabrata*. *C. auris* strains were obtained from the U.S. Centers for Disease Control and Prevention [14]. Fungal cell stocks were stored at −80 °C and propagated by streaking onto yeast extract peptone dextrose (YPD) agar plates and incubated overnight at 30 °C. From these overnight cultures, a loopful of cells was inoculated into flasks (150 mL) containing 25 mL of YPD liquid media and grown overnight for 14–16 h at 30 °C with shaking (180 rpm). Under these conditions, both *Candida* and *Cryptococcus* grow as budding yeast.

### 2.2. Antifungal Susceptibility Testing

Fluconazole (Flu, Pfizer Central Research, Sandwich, UK), amphotericin B (AmB, Bristol-Myers-Squibb, Woerden, The Netherlands), and caspofungin (Cas, Sigma-Aldrich, St. Louis, MO, USA) were used in this study. In vitro antifungal susceptibility testing against planktonic yeast cells was carried out using CLSI M27 broth microdilution method with minor modifications [15]. The range of concentrations tested were 0.25–128, 0.015–8.0, and 0.031–16.0 µg/mL for Flu, AmB, and Cas, respectively. The microdilution plates were incubated for 24 h at 37 °C for *Candida* species. Cell growth (optical density) was measured using Synergy H1 microplate reader (BioTek, Winooski, VT, USA) and the minimum inhibitory concentrations (MIC_50_ and MIC_90_) were defined as the lowest concentration of a drug that significantly reduced growth (>50% and >90% respectively), compared to the growth of a drug-free control. Determination of MIC of the identified compounds against *Cryptococcus* was also conducted following the CLSI M27-A3 protocol with minor modifications. Serial twofold dilutions of all drugs were prepared in RPMI 1640 medium buffered with MOPS. The final drug concentrations in 96-well microtiter plates were 0.1–20 µg/mL. After 48 h incubation, viable *Cryptococcus* yeast was determined by colony counting. The percentage growth inhibition was calculated by (CFU of control − CFU of treated yeast)/(CFU of control) × 100%.

### 2.3. Cell Labeling

Yeast cells were harvested from overnight cultures, washed, and resuspended in RPMI 1604 with glutamine, but without phenol red. Approximately 1 × 10^6^ yeast cells in 50 µL were aliquotted in each well of 96-well microtiter round-bottom plates, and separately incubated with a drug at a fixed concentration of 10–20 µM (1: 100 dilution) at 37 ℃, 5 % CO_2_ for 8 h for *Candida* spp., and 48 h for *C. neoformans*. Cells separately incubated with 2% DMSO (drug vehicle) and 1 µg/mL AmB in RPMI 1640 were used as negative and positive controls, respectively. Fungal cells were washed once with PBS pH 8.1 and stained with a master mixture of calcofluor white (0.00001% we/vol; Sigma, St. Louis, MO, USA) and FM4-64FX (1:20 dilution; Invitrogen, Cat# F34653) for 30 m at 37 °C. Following incubation, cells were washed twice with PBS before being fixed in 30 µL of 2% paraformaldehyde (PFA). Fold change of mean of fluorescent intensity (MFI) for each drug was calculated using the following formula: [(MFIx for drug X − MFI_DMSO_)/MFI_DMSO_].

### 2.4. Imaging Flow Data Acquisition

Cells were analyzed using a 5-laser 12-channel Amnis ImageStreamX MKII image flow cytometer equipped with an autosampler. Data were acquired for 3 min (~5000 cells per sample) using INSPIRE Software. Three lasers (405-, 642-, and 785-nm) were used to excite the cell-labeled fluorochrome. Cell images were acquired using a 40× or 60× objective lens in the corresponding channels. Cells incubated with each fluorochrome individually were used to establish a color compensation matrix. The acquired raw data for each sample were then analyzed using IDEAS^®^ 6.2 software, and a gating template was created. All samples were batch-analyzed using the same color compensation matrices and data analysis templates for *Candida* and *Cryptococcus*. Quality control for plate-to-plate variation was conducted by comparison of the output values for control and AmB-treated cells to the day-one values when the protocol was established. Variability of positive and negative control values were within ±10%.

### 2.5. Chemical Library and Drug Treatment for Imaging Flow Analysis

The Prestwick Chemical library (Prestwick Chemical, Graffenstaden, France) contains approximately 1280 chemically and pharmacologically diverse compounds, mostly off-patent approved drugs used for the treatment of a variety of diseases, and approximately 5% of other drugs at different stages of development, all with known bioavailability and safety in humans. The compounds in the library are provided in 96-well microtiter plates as 10 mM solutions in dimethyl sulfoxide (DMSO). Working stock plates were prepared at 1 mM in 100% DMSO.

### 2.6. Data Analysis

The feature values (cell length, area, and fluorochrome intensity) of each individual cell in the gated population of a sample were exported to Excel and GraphPad Prism 8 for subsequent data analysis and statistical assessment, respectively. One-way ANOVA was used for multiple comparisons. A *p* valve ≤ 0.05 was considered statistically significant.

## 3. Results

### 3.1. Optimization of Culture Conditions and FCP Analysis for C. albicans

Yeast cells were grown in buffered RPMI-1640 medium at 37 °C with 5% CO_2_ in order to induce the initiation of the yeast-to-hyphae transition. We established a multi-parameter profiling method to examine the cellular states and morphology of *C. albicans* using a 12-channel imaging flow cytometer. At least 5000 single cell images were obtained for each sample in 3 min and subsequently analyzed using IDEAS^®^ software. The gating strategies shown in Figure 1 were used to separate the cells of interest from the out-of-focus cells, debris, and aggregates. Briefly, gates were drawn to select in-focus cells using gradient root mean square (RMS) method, which measured the sharpness quality of an image through the enumeration of pixel values (>46). Subsequently, plots of area versus intensity of the side scatter channel (SSC) were utilized to separate the cells of interest from the debris and aggregates. Furthermore, plots for cell width versus aspect ratio (minor axis/major axis) were employed to further select singlets that had a width within the normal range of these fungal cells (5–15 pixels represent 2.5–7.5 µm in diameter). A total of at least 2000 images of singlets were included for statistical analysis. Finally, subpopulations of yeast cells, elongated yeasts, and hyphae of *Candida* cells were defined based on cell size (area) and aspect ratio as shown in the Figure 1.

*C. albicans* cells were harvested at 0, 2, 4, 6, and 8 h after inoculation to establish cytological profiles of fungal growth. Cells were stained with calcofluor white (CFW) and a fixable lipophilic styryl dye (FM4-64FX) which bind to the cell wall (chitin) and vacuolar membrane system, respectively [16,17]. These two dyes were utilized to detect changes in chitin content, cell wall integrity, lipid content, and cell permeability. Cells were analyzed by using the fungal cytological profiling as described above. Over time, the yeast became elongated to form budding cells or germ-tube initials at ~2 h after inoculation (Figure 2A). They progressively transitioned to septated hyphae with increasing length over the course of the 8 h incubation. After 10 h, *C. albicans* began to form branching hyphae that could clog the cytometer; therefore, the growing time was set to be 8 h (Figure 2A,B). After the yeast–hyphae transition, the mean fluorescent intensity (MFI) of FM4-64FX significantly decreased († *p* < 0.05 one-way ANOVA test) and it became consistent after 6–8 h (Figure 2C). The MFI of CFW was initially increased during the first 2 h, while the hyphae elongated, and the CFW-labeled concentrate on the septa and total intensity was decreased (Figure 2D). Furthermore, the CFW fluorochrome intensity per cell length unit became consistent after 6 h when over 95% of yeast cells had converted to hyphae (Figure 2E). Collectively, a set of profiling data was established for size (e.g., length, width, and area), shape (e.g., aspect ratio), texture (e.g., gradient RMS and contrast), signal intensity (e.g., CFW and FM4-64FX fluorochrome intensity), and any combined features (e.g., fluorescent intensity/length) of *Candida* cells under this culture condition.

### 3.2. Validating C. albicans FCP after Exposure to Clinical Antifungal Drugs

We first tested our profiling method using *C. albicans* SC5314, which was separately incubated for 8 h with fluconazole (Flu), amphotericin B (AmB), and caspofungin (Cas) at 8, 1, and 0.5 µg/mL, respectively. The testing concentration of Flu was slightly lower compared to its MIC_50_ (16 µg/mL) of this strain, while those of AmB and Cas were higher compared to their respective MIC_50_s, 0.125 and 0.25 µg/mL, respectively. These concentrations were used to validate whether the newly established FCP could distinguish between effective and non-effective drugs. Representative images of the major population of the untreated control and treated samples are illustrated in Figure 3A showing bright field (BF), calcofluor-white-stained cells (CFW), FM4-64FX (FM4), and merged images from the two fluorochrome channels (CFW+FM4). The overall changes in cell shape (yeasts-to-hyphae transition), size (length), and integrity CFW /length by each drug treatment were analyzed (Figure 3B–E). Notably, at the concentration used, fluconazole treatment did not prevent the yeast-to-hyphae transition but the length of hyphae was significantly reduced compared to the untreated control († *p* < 0.0001). Furthermore, CFW and FM4-64FX fluorescence densities were significantly increased (* *p* < 0.0001), indicating that cell wall integrity and membrane permeability were impeded by fluconazole treatment. Predictably, there was a significantly reduced percentage of hyphae in AmB- and Cas-treated cells (Figure 3B,C; † *p* < 0.0001). Furthermore, many other FCP parameters were significantly altered in both AmB- and Cas-treated *C. albicans* cells, including the percentage of hyphae, cell length, and density of CFW and FM4-64FX fluorescent intensity compared to the untreated cells (Figure 3B–E). AmB-treated cells had a significant decrease in CFW and FM4-64FX intensity, while Cas-treated cells showed increased intensity to these dyes, suggesting susceptibility to both drugs.

### 3.3. The Newly Established FCP Method Is Useful for Profiling Fungal Cellular States

To further validate the FCP method, four strains of clinically-relevant *Candida* species including *C. auris* (strains 0384 and 0390), *C. glabrata,* and *C. lusitaniae* were tested. These three *Candida* species grew as oval-shaped yeast in buffered RPMI-1640 at 37 °C with 5% CO_2_ for 8 h [14,18]. Results showed that both *C. auris* strains were resistant to Flu (MIC > 128 µg/mL) and their CFW and FM4-64FX intensity were not altered at the testing concentration (8 µg/mL). In contrast, those parameters were changed for all tested strains that were incubated with AmB and Cas at their corresponding MIC_50_ (Figure 4A–D). *C. glabrata* and *C. lusitaniae* were susceptible to all three antifungals at the drug concentrations tested, as indicated by their altered FCPs (Figure 4E–H). These results validate that our FCP method can measure a set of cell parameters for predicting drug susceptibility of *Candida albicans* as well as other *Candida* species that do not undergo the yeast–hyphae transition.

### 3.4. Profiling Cytological States of C. neoformans after Exposure to Clinical Antifungal Drugs

We further extended this FCP method to the *C. neoformans* H99 strain that was also cultured in the same buffered RPMI-1640 medium in the absence or presence of antifungal drugs. H99 yeast cells were stained with CFW and FM4-64FX and their images were acquired using the same acquisition program for *Candida* species. The gating strategy is illustrated in Figure 5. A set of feature parameters for the H99 yeasts were established for size, shape, texture, signal intensity, and any combined characteristics. These values for untreated H99 yeast did not change for the period of a 48 h incubation; therefore, all subsequent experiments were performed using the 48 h incubation condition. Images of H99 yeast cells after incubation in the presence of a sub-inhibitory concentration of Flu (8 µg/mL) were comparable to controls (Figure 6A,B). Notably, H99 FCPs were visibly altered after incubation with AmB and Cas at their MICs (0.5 and 16 µg/mL, respectively) (Figure 6C,D). Digitalized values of cell size (area) and fluorescence intensity of CFW were significantly reduced for AmB- and Cas-treated yeast compared to controls (Figure 6E,F). Fluorescence intensity of FM4-64FX was significantly reduced for AmB-treated yeast, while it was elevated for Cas-treated cells compared to control (Figure 6G). These data suggest that the FCP method is sensitive, quantitative, and objective for detecting cellular impact of antifungal drugs.

### 3.5. Potential Application of the FCP Method for Screening a Drug Library to Identify Compounds with Antifungal Activity

Once the FCP method was fully validated as a sensitive technique for evaluating the effects of antifungal drug treatment on individual cells and cell populations, we investigated if this methodology could also be potentially utilized as the basis for a screening methodology for antifungal drug discovery. As a proof of concept, we used this FCP platform to screen the Prestwick library consisting of 1280 off-patent drugs in order to identify those that could alter the cellular profiles of *C. neoformans*. Yeast cells were seeded in round-bottom 96-well plates, individually treated with each drug in the Prestwick library at a fixed concentration of 20 µM for 48 h. A scatter plot of the Log_2_ (fold changes of CFW intensity) versus Log_2_ (fold change of FM4-64FX intensity) for the 1280 drugs is illustrated in Figure 7A. A total of 17 drugs that caused a change of over twofold (outside the boxed area) in CFW alone, FM4-64FX alone, or both were identified, resulting in a 1.3% initial hit rate. The antifungal activity of 12 of these drugs against *C. neoformans* H99 yeast was confirmed using CLSI methodologies by a broth dilution antifungal susceptibility test (Figure 7B). These compounds consist of four dopamine antagonists for antipsychotic treatment, three anti-inflammatory drugs, two antihistamine drugs, one anti-tapeworm drug, one selective estrogen receptor modulator, and one anesthetic drug (Table 1). The four antipsychotic drugs are all belong to the phenothiazine class and their MIC_50_ against *C. neoformans* were further determined (Table 1) to range from 0.43 to 11.12 µg/mL.

Butts et al. [19] have previously reported 31 hits using an adenylate kinase screen of the Prestwick library against the same serotype H99 strain of *C. neoformans* used in this study. Among those 31 compounds, only three, Trioridazine, Trifluoperazine, and Tamoxifen, were identified from our screen showing the alteration of *C. neoformans* FCPs. These three compounds and additional seven of the current twelve hits have been previously reported and/or patented to have antifungal activity (Table 1). Perphenazine and oxethazaine are two newly identified drugs in this study (Table 1). Thus, results from this pilot study demonstrate the potential applicability of this newly established imaging flow cytometry for antifungal drug discovery.

**Table 1 jof-09-00722-t001:** Identified drugs from the Prestwick Chemical Library against *C. neoformans*.

Drug Type	Drug Name	MFI Fold Change		MICs (µg/mL)	Butts et al. [19] Reported Hits	Reported Drugs against Other Fungal Species and Cited References	
		CFW	FM4-64				
Phenothiazine antipsychotic	Thioridazine	−3.79	2.27	1.77 (50%)	Y	*Paracoccidioides Cryptococcus*	[20] [19]
	Chlorpromazine	−2.49	3.72	2.28 (50%)	N	*Aspergillus* *Candida* *Cryptococcus*	[21] [22] [23]
	Trifluoperazine	−2.68	3.99	3.36 (50%)	Y	*Aspergillus* *Candida* *Cryptococcus*	[22] [23] [24]
	Perphenazine	−1.40	2.87	1.20 (50%) 3.50 (90%)	N	*Cryptococcus*	This study
Anti-inflammatory	Auranofin	−4.57	−2.05	2.41 (50%)	N	*Aspergillus* *Blastomyces* Candida *Cryptococcus*	[25] [26] [27]
	Ebselen	−5.40	−3.95	0.88 (50%)	N	*Aspergillus* *Candida* *Cryptococcus*	[28] [29] [26]
	Nimesulide	−1.38	−2.68	9.83 (50%)	N	*Aspergillus,* *Candida* *Cryptococcus*	[30]
Antihistamine	Astemizole	1.66	2.19	0.43 (50%)	N	*Cryptococcus*	[31]
	Terfenadine	1.63	2.56	7.78 (50%)	N	*Candida*	[32]
Antihelmintic	Niclosamide	−2.16	−1.74	6.42 (50%)	N	*Candida*	[33]
Antineoplastic	Tamoxifen	−1.14	2.90	1.21 (50%)	Y	*Candida* *Cryptococcus*	[34] [35]
Anesthetic	Oxethazaine	−1.87	3.05	11.1 (50%) >20.0 (90%)	N	*Cryptococcus*	This study

## 4. Discussion

Imaging flow cytometry analyzes fluorescence signals in the context of morphological characteristics and spatial distributions that are essential for measuring many biological functions including nuclear translocation [36], co-localization using “similarity” features [37], calcium signaling at the organelle level [38], cell division [39], yeast cell cycle [40], and drug screening against malaria [41], amongst other analyses [42,43]. These assays would not be possible using traditional flow cytometry (lack of spatial information) or conventional imaging techniques (low throughput and poor quantitation). Here, we have applied this technique to the fungal cytological profiling of antifungal drug effects, and we have also described its potential application as a screening methodology in the search for compounds with novel antifungal activity.

First, we established FCPs for three first-line clinically-used antifungals, each with a distinct mode of action. Fluconazole works by inhibiting the synthesis of ergosterol in the cell membrane of fungal species, acting as a fungistatic drug, but has the highest prevalence of resistance of the antifungals, particularly in *Candida* spp. and *C. neoformans* [44]. Amphotericin B displays the most broad-spectrum activity among all of the clinically-used antifungals. AmB binds to ergosterol and disrupts the fungal cell membrane barrier, leading to a loss of intracellular components. Most fungal isolates are highly susceptible to this polyene, but *C. lusitaniae* strains can show resistance, possibly due to a lower ergosterol content in comparison to other species of fungi [45,46]. Caspofungin is one of the more recently discovered antifungals, acting on β-(1,3)-glucan synthesis in the fungal cell wall. However, it has recently been reported that some *C. glabrata* strains have shown Cas-resistance in vivo, but are susceptible in vitro [47]. Furthermore, *Candida auris* and *Cryptococcus neoformans* are notoriously resistant to many antifungal agents [29,48,49,50,51,52]. In this study, all Cas-treated fungal cells showed an increase in chitin binding with CFW, as well as elevated intensity of FM4-64FX, which may be due to a decrease in the structural β-(1,3)-glucan cell wall, allowing more of the fluorochrome dyes to bind or enter the cells. Comparison of the FCPs of cells treated with these three known antifungals served as the basis of subsequent screening experiments to identify compounds that can cause similar changes in the cytological profiles of fungal pathogens, and therefore may represent potential novel antifungals.

The method developed here uses a simple and inexpensive 96-well plate-based platform, in which treatment and staining were performed and standardized across several fungal species. The IFC allows for the high-content analyses of up to 1000 cells per second. Gating strategies were established in the software for evaluating several fungal morphologies, allowing for the rapid analysis of numerous samples. Our fungal cytological profiling platform is easy-to-use and provides an objective statistical analysis. The labeled cells could be fixed and stored at 4 °C in the dark for a week without losing fluorochrome intensity and the entire automated imaging acquisition procedure takes ~4 h per plate. We posited that all these characteristics make this methodology potentially desirable as a novel drug-screening platform, in which a rapid and efficient assessment is necessary for identifying hits. Thus, once the FCP platform was fully established we wanted to explore its potential application as an antifungal drug-screening platform, despite the fact that it is not a high-throughput method compared to other enzymatic and metabolic screening methodologies. However, IFC provides visual confirmation to assess the drug-mediated alteration of fungal characteristics, e.g., yeast-mycelium transition, morphological alteration, cell wall integrity, and membrane damage.

As a proof of concept of its applicability in drug screening, we use this methodology to screen the Prestwick library against *C. neoformans*. Twelve of the seventeen initially identified hits were confirmed to inhibit the growth of *Cryptococcus* yeast in vitro with an MIC_50_ between 0.4–11.1 µg/mL (see Table 1). Chlorpromazine, thioridazine, and trifluoperazine are originally classified as phenothiazine antipsychotic drugs [19,20,21,22,23,24], which may represent a potentially useful scaffold for the further optimization of new antifungal drugs. Auranofin, ebselen, and nimesulide are anti-inflammatory compounds [18,25,26,27,28,30]. Astmizole and terfenadine are antihistamine compounds [31,32]. Additionally, niclosamide, and tamoxifen are anti-helminthic and antineoplastic drugs [33,34,35,53]. Interestingly, tamoxifen has been through a Phase II clinical trial for the treatment of *Cryptococcus* infection [54].

In conclusion, our newly developed image-profiling method represents a fast, specific, and highly reproducible technique for assessing the effects of drugs on the cytological profiles of fungi. This new method has been validated amongst several pathogenic fungi, and may also have potential utility in screening applications for the identification of compounds with novel antifungal activity. This fungal cytological-profiling technology could also be a powerful tool for identifying the drug’s mode of action for the development of novel antifungals.

## Figures and Tables

**Figure 1 jof-09-00722-f001:**
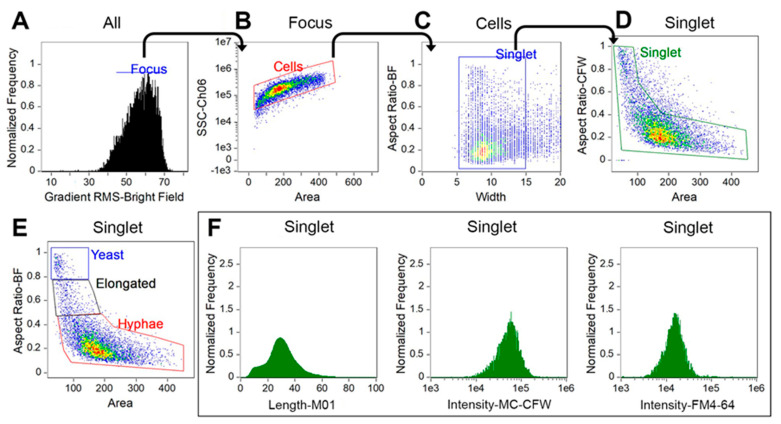
Gating strategy for profiling *Candida albicans*. Focused fungal cells were identified using gradient root mean square (RMS) method (**A**), then separated from aggregates and debris on the basis of area and side-scatter area (SSC-Ch06) (**B**). The remaining debris and aggregates were further filtered out based on width and area versus aspect ratio (**C**,**D**). Populations were then grouped by morphology and separated into three subpopulations, including yeast, germ tube-forming cells (i.e., elongated), and hyphae based on aspect ratio and area (**E**). Representative histographs of length, fluorochrome intensity of calcofluor white, and FM4™4-64 are shown (**F**). Other parameters such as cell area, circularity, and any meaningful features of the assays can also be created for the target subpopulations of the gated singlet cells.

**Figure 2 jof-09-00722-f002:**
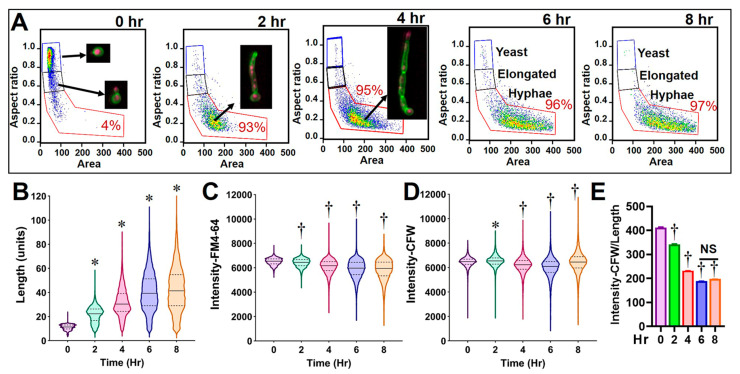
Profiling *Candida albicans* cells using imaging flow cytometry analysis. *Candida* cells were grown in RPMI-1640 medium at 37 °C with 5% CO_2_ for a period of 8 h and stained with calcofluor white (CFW) plus FM4-64 for FCP analysis. Gating strategy for focused and single cells is shown in Figure 1. Subpopulations of yeast, elongated cells (budding or germ tube-formatting), and hyphae in 0–8 h cultures were defined from gated single cell population by dotted plots using area versus aspect ratio (**A**). Representative images of each subpopulation are shown in the inserts. At 2 h after inoculation, 93 ± 2.2% of yeast grew into hyphae. Percentage of hypha subpopulation is indicated in each analysis panels. Data of IDEAS^®^-software-defined features for each sample (>4000 cells) were exported and analyzed for cell length (**B**), and fluorochrome intensity of FM4-64 (**C**) and CFW (**D**) using GraphPad (violin plots with median and quartile lines). Data for newly created feature (i.e., intensity of CFW/length) were also exported and plotted (**E**). * *p* < 0.05 and † *p* < 0.05 represent significant increase or decrease, respectively, between the corresponding feature at indicated time points and 0 h control samples, by one-way ANOVA test.

**Figure 3 jof-09-00722-f003:**
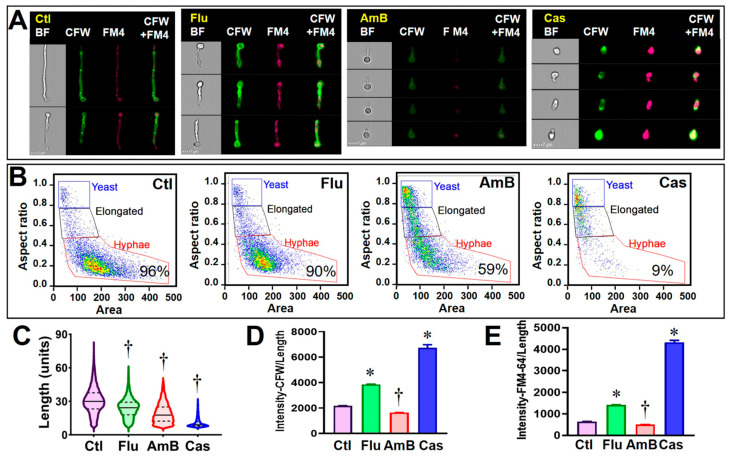
Cytological profiling of *C. albicans* following exposure to clinical antifungal drugs. *Yeast* grown in RPMI-1640 medium was treated with fluconazole (Flu), amphotericin B (AmB), and caspofungin (Cas) for 8 h, or left untreated as control (Ctl). Calcofluor white (CFW) and FM™4-64 (FM4) were then mixed to stain fungal cell wall and vacuolar membrane. At least 2000 focused images were analyzed for each sample. Representative images of treated and control (untreated) *C. albicans* cells are shown in (Panel (**A**)). The percentage of hyphal subpopulation under each treatment condition and control is indicated in each plot (**B**). Alteration of fungal growth by drug treatment was plotted with histogram of hyphae length (**C**). Alteration of cell wall integrity and membrane permeability by drug treatment was plotted as histograms of CFW and FM™4-64 binding intensity normalized with cell length (**D**,**E**). * and † indicate statistically significant increase and decrease between treatment and control groups, respectively (one-way ANOVA test, *p* < 0.01).

**Figure 4 jof-09-00722-f004:**
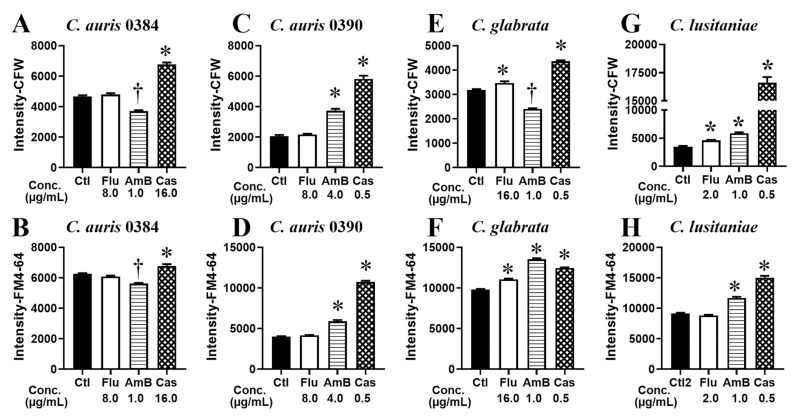
Cytological profiling of *Candida* spp. following exposure to clinical antifungal drugs. *C. auris* (0384 and 0390 strains), *C. glabrata,* and *C. lusitaniae* were treated with fluconazole (Flu), amphotericin B (AmB), and caspofungin (Cas) for 8 h, or left untreated as control (Ctl). Fluorochrome intensity of calcofluor white (**A**,**C**,**E**,**G**) and FM4-64 (**B**,**D**,**F**,**H**) for each *Candida* species was plotted for each drug treatment. The relative fluorescence intensities (median ± SEM) for each treatment group compared to control were graphed. * and † indicate statistically significant increase and decrease between treatment and control groups, respectively (one-way ANOVA test, *p* < 0.01).

**Figure 5 jof-09-00722-f005:**
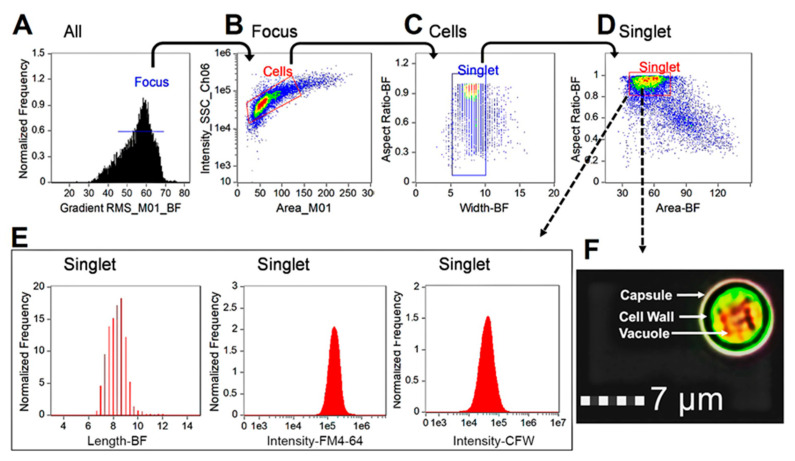
Gating strategy for *Cryptococcus neoformans*. Focused yeast (H99 strain) were identified by RMS (**A**). Yeast singlets were further selected by separation from aggregates and debris on the basis of two plots using area versus side scatter channel (SSC-Ch06) and width versus aspect ratio (**B**,**C**). Finally, the singlets were gated on the dotted plot using area versus aspect ratio (**D**). IDEAS^®^-software-defined or newly created features such as cell length and fluorochrome intensity of FM4-64 and CFW for each sample were analyzed and plotted (**E**). A typical H99 cell in the defined yeast singlet population was shown (**F**).

**Figure 6 jof-09-00722-f006:**
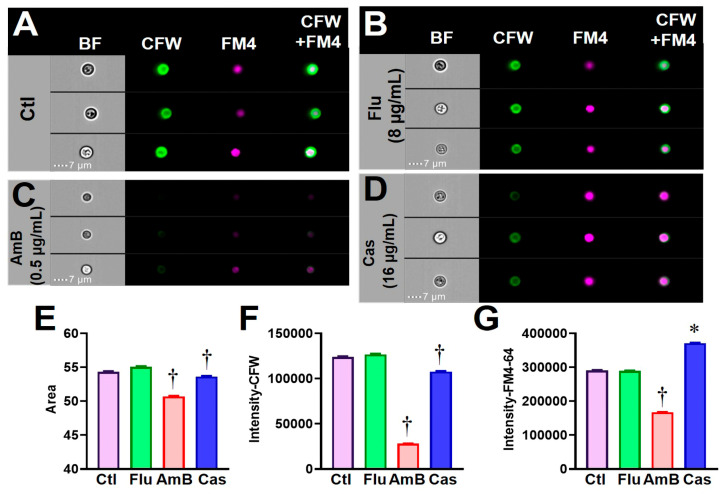
Cytological profiling of *Cryptococcus neoformans* following exposure to clinically-used antifungal drugs. Yeast grown in RPMI-1640 medium was treated with a sub-inhibitory concentration of fluconazole (Flu), or high concentrations of amphotericin B (AmB) and caspofungin (Cas) for 48 h, or left untreated as an untreated control (Ctl). At least 2000 cell images fulfilling the gating criteria were analyzed for each sample. Representative images of antifungal-treated and untreated (control) *C. neoformans* cells are shown (**A**–**D**). The cell size (area) and relative fluorescence intensities (median ± SEM) for each treatment group compared to control are graphed (**E**–**G**). * and † indicate statistically significant increase and decrease between treatment and control groups, respectively (one-way ANOVA, *p* < 0.01).

**Figure 7 jof-09-00722-f007:**
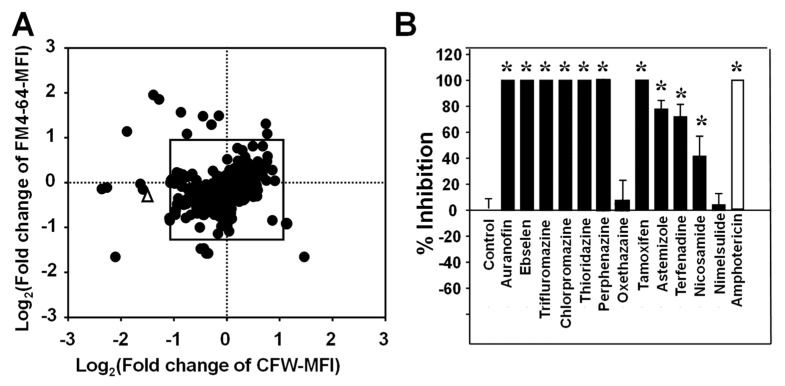
Screening of the Prestwick Library to identify potential antifungals against *C. neoformans* using fungal cytological profiling. (**A**) *C. neoformans* yeast cells (1 × 10^6^) were seeded in each well of a round-bottom 96-well plate. Cells within each well were incubated with individual compounds in this repurposing library, and stained with a mixture of calcofluor white (CFW) and FM™4-64. Treatment with AmB at 1 µg/mL was used as a positive control, whereas the CFW and FM™4-64 mean fluorescence intensity (MFI) of the untreated samples in wells of the same plate were used as negative controls (100% growth). At least 2000 focused cell images from each compound-treated sample were analyzed. A scatter plot of the fold changes of CFW versus FM™4-64 intensity for the 1280 drugs (solid circules) and mean valuve of AmB from each plate (empty triangle) was drawn to identify potential antifungals. The antifungal activity against *C. neoformans* for selected drugs from the initial screening using a growth inhibition assay at a concentration of 20 µM of each tested compound (black bars) and 1 µg/mL AmB (white bar) (**B**). Further determination of MIC for each selected drug from (**B**) was conducted and data were shown on Table 1. * indicate statistically significant differences between drug treated and untreated control groups (Student’s *t*-test, *p* < 0.01).

## Data Availability

The data presented in this study are available on request from the corresponding author. The data are not publicly available due to privacy reason.

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
