# Peer review of "Development of an Imaging Flow Cytometry Method for Fungal Cytological Profiling and Its Potential Application in Antifungal Drug Development"

_jof, 2023, doi:10.3390/jof9070722_

Round 1

Reviewer 1 Report

Journal of Fungi - jof-1996473

This manuscript presents a flow-cytometry based imaging assay which uses side scatter and two fluorescent dyes to generate a profile of cellular phenotypes that can be used to infer membrane texture and permeability, cell wall changes, cell size and length. This assay is used with clinical antifungals against Candida species and C. neoformans. Though the profiling of fungal cells treated with antifungal compounds can successfully identify physical changes associated with treatments, this assay has not been validated enough for use as a high throughput screening assay. However, the assay can rapidly generate a large amount of data pertaining to cellular health and has great potential to be used as a secondary screening method following a primary drug screen (with some additional validation studies) which could help to eliminate toxic compounds and direct investigators towards pathways of interest (cell wall, cell membrane, hyphal induction, cell cycle, etc.) during mechanism of action studies. 

1.     I recommend the authors reframe the use of this assay from a high throughput primary screening assay to a secondary screening assay to characterize candidate compounds.

2.     As a secondary screening assay, this technique could be very powerful in characterization of novel antifungal compounds; importantly these profiles might help rapidly eliminate membrane disrupting compounds. This assay could be tested against a large number of general membrane-disrupting disinfectants to see if there is a common FCP profile with these treatments which can then be used to identify them before ever performing any toxicity assays.

3.     The cytological profiling assay has not been thoroughly validated. Line 186 “These concentrations were used to validate whether the newly established FCP could distinguish effective and non-effective drugs” This claim is made however no non-effective drugs are tested. The assay should be tested against a variety of effective and non-effective drug treatments and stresses to determine how often compounds can have significant effects on FCP parameters without growth inhibitory effects and how often drugs with potent activity do not affect FCP parameters. It is also important to know similar drug affect FCP profiles; do all azoles induce the same profile? Echinocandins? Do the changes in magnitude of the parameters correlate with antifungal activity or do FCP profiles only give information about the presence or absence of activity, and not the potency? Using structural analogs that have a range of activity from very potent to no activity would be an excellent way to approach this question.

4.     The authors screened the Prestwick library as a proof of concept for their assay. This library has been screened at least once against C. neoformans and thus there are several known compounds with antifungal activity in this library. There is no mention of previous screening results and how their results compare. Several compounds have known activity in this library that are not mentioned here (See https://www.ncbi.nlm.nih.gov/pmc/articles/PMC3571299/pdf/zek278.pdf) . In these early validation stages, it is very important to determine whether these hits did not come up due to a difference in screening methodologies (growth conditions, strain, etc.) or if the FCP profiling missed these compounds because they have antifungal activity but do not change the FCP parameters.

5.     Some grammar and typos to address:

a.     Line 94 “optimal density” ïƒ  optical density

b.     Line 140 Prim8 ïƒ  Prism8

c.     Line 192 “notably, flu-treated yeast..” should read “Notably, fluconazole treatment did not prevent the yeast-to-hyphae transition”

d.     Line 209 “buffered in PRMI” ïƒ  RPMI

If the assay will be presented as a primary screening method, the following experiments/comments will also need to be addressed:

6.     Related to comments 4 and 5, what are the false positive and negative hit rates of the assay? How do these hold up against a library that is less biased towards compounds that have biological activity? How many compounds affect FCP but not growth and vice versa?

7.     What is the well-to-well and day-to-day variability of this assay? A Z prime score should be generated on independent days to assess the variability between assays and between wells of an assay; these scores can help guide whether an assay is robust enough for high throughput screening and should be performed rigorously before moving into screening.

8.     It is unclear what the throughput of this assay is; the authors mention (line 307) that it takes 1-2 days per plate; does this mean a single plate can be processed per day? Can this assay be formatted in 384-well plates? This seems like a low to moderate throughput assay that would not be feasible for use in high throughput screening.

Author Response

We thank the academic editor and the reviewers for their constructive suggestions and we appreciate the opportunity to address the concerns. We have revised the manuscript with point-by-point responses as described below. The manuscript text is also revised accordingly.  A clean copy of the revised manuscript and one with track changes are submitted.  A copy of response letter is attached.

Reviewer 2 Report

The article “Development of a High Throughput Imaging Flow Cytometry Method for Fungal Cytological Profiling and Antifungal Drug Discovery’ describes a new methodology for antifungal drug screening. It showed an interesting method that can be more sensitive than others that used optical density or colorimetric assays.

During the analysis, some questions occurred:

1-      Does that mean small molecules? Because the authors emphasize the small molecules in the manuscript’s aim. It could be added to the introduction.

2- the introduction should describe false-positive and false-negative results of optical density or colorimetric methods.

3-    Why the authors used only the MIC50 and not MIC90, considering the importance of both in drug screening?

4-    Why did the authors not use a more sensitive methodology, like a live-dead viability assay? Some drugs can be fungistatic, and the liver cells will not grow in a culture medium. 

5-    How many samples did the authors read? It is not described in figures legends.

6-    What occurs when different concentrations of one drug range from lower to higher MIC50? How is the changing of fluorescence pattern? The authors should show a known drug's MIC determination with this methodology.

7-    Page 7 – line 209 – RPMI and not PRMI

8-    The description of several advantages included the simple and inexpensive method but the need for specific equipment for this methodology. It should be better explained.

9-    What cell characteristic is the best to be used? In the discussion, the authors described the characteristic they used in the manuscript, but I think they would be more assertive in this message. 

10- They should show the same assay using colorimetric assays to confirm the sensibility of the new methodology.

Author Response

We thank the Reviewer#2 for the constructive suggestions. We have revised the manuscript with point-by-point responses as described in the attached file. The manuscript text is also revised accordingly.  A clean copy of the revised manuscript and one with track changes are submitted.  

Round 2

Reviewer 1 Report

The revisions of this paper do not adequately address my concerns about the validity of this method as a drug screening platform. 

1. The authors have included a new statement in the introduction Line 61 " A secondary assay is often applied to remove false-positive hits due to chemical impurities...Incorporation of a phenotypic screen may provide valuable function to remove false positives and rescue false negative compounds" There are two issues with this statement. First, the impurities in compounds that result in false activity in an assay will be present when using this screen and are just as likely to alter cellular profiles due to the stresses the authors note (non specific binding, aggregation, redox reactions, etc). Nothing about this profiling assay makes it immune to the effects of compound degradation or impurities. Second, though they say the advantage of this assay is to improve the false hits generated in traditional screens, the pilot screen the authors perform has a lot of false negatives. They cite Butts et al who used an AK screen to identify 31 compounds; Butts et al tested the activity of all the hits they identified. However in the screen presented using IFC, many of the hits that had a  MIC against C. neoformans between 4-8 identified by Butts et al (Sertraline, Benzethonium, Suluctodil, Perhexiline) were not identified in the IFC screen. Thus all of these hits identified in previous screens that were confirmed to have activity would be considered false negatives of this screen. 

The authors still claim (Line 192) that they are testing whether the assay can distinguish from effective and non effective drugs, however the figure still only utilizes DMSO control or clinical antifungals. The best test of an assay to determine true antifungal activity from off target effects of a compound is to use analogs of the same chemical structure with different activities. For example there are many compounds in the phenothiazine class of molecules that have antifungal activity and those that don't; how do each of these alter the FCP and are there parameters that change with antifungal activity and those that don't? It still remains that the authors did not include any non-effective drugs (meaning no activity at any concentration, not just a lower concentration of a drug with activity) while validating their screening assay. The response that "most of the Prestwick drugs were non effective"  is also invalided by the fact that this IFC screen failed to identify several drugs in this library that already have been shown to have reasonable activity against C. neoformans. 

My suggestion remains that this manuscript should be presented as a secondary screening assay and not a primary antifungal drug screen. The assay is low throughput at best and and has not been adequately validated for drug screening.  This is nice work and the assay presented here has great value but it still lacks the validation and rigor to be considered a drug screening assay. 

Author Response

Dear Reviewer #1:

The manuscript has been revised to further de-emphasize screening.  A point-by-point response letter is attached below.  Thank you for your constructive suggestions.

Round 3

Reviewer 1 Report

This revision addresses my concerns about this assay as a primary screening platform; I appreciate my comments being taken into consideration. This work is well done and will add a lot of value to drug development.